# Laparoscopic Robotic Surgery: Current Perspective and Future Directions

**Sally Kathryn Longmore** [1,*] , **Ganesh Naik** [1] and **Gaetano D. Gargiulo** [1,2,3,4]

1   MARCS Institute for Brain, Behaviour and Development, Western Sydney University, Milperra, NSW 2560, Australia; ganesh.naik@westernsydney.edu.au (G.N.); g.gargiulo@westernsydney.edu.au (G.D.G.)
2   School of Engineering, Western Sydney University, Kingswood, NSW 2747, Australia
3   Translational Health Research Institute, Western Sydney University, Campbelltown, NSW 2560, Australia
4   Ingham Institute, Liverpool, NSW 2170, Australia
*   Correspondence: s.longmore@westernsydney.edu.au

**Abstract:** Just as laparoscopic surgery provided a giant leap in safety and recovery for patients over open surgery methods, robotic-assisted surgery (RAS) is doing the same to laparoscopic surgery. The first laparoscopic-RAS systems to be commercialized were the Intuitive Surgical, Inc. (Sunnyvale, CA, USA) da Vinci and the Computer Motion Zeus. These systems were similar in many aspects, which led to a patent dispute between the two companies. Before the dispute was settled in court, Intuitive Surgical bought Computer Motion, and thus owned critical patents for laparoscopic-RAS. Recently, the patents held by Intuitive Surgical have begun to expire, leading to many new laparoscopic-RAS systems being developed and entering the market. In this study, we review the newly commercialized and prototype laparoscopic-RAS systems. We compare the features of the imaging and display technology, surgeons console and patient cart of the reviewed RAS systems. We also briefly discuss the future directions of laparoscopic-RAS surgery. With new laparoscopic-RAS systems now commercially available we should see RAS being adopted more widely in surgical interventions and costs of procedures using RAS to decrease in the near future.

**Keywords:** robotic-assisted surgery; laparoscopic; 3D stereoscopic imaging; haptic feedback; tremor removal; patents; hand controllers; end effectors

## 1. Introduction

In the year 1986, a team using a modified UNIMATION PUMA 200 programmable industrial robotic arm performed the very first robotic assisted surgery (RAS). The surgical procedure used the PUMA 200 robot to obtain a biopsy from a patient with a suspected brain lesion [1,2]. Since this first successful use of a robot to assist in a surgical procedure, several RAS systems have been developed, but only few of those systems have been commercialized.

The first two laparoscopic RAS systems to be commercialized were the Intuitive Surgical Inc. (Sunnyvale, CA, USA) da Vinci and the Computer Motion Zeus. The da Vinci RAS system was the first to receive US Food and Drug Administration (FDA) approval in 2000, while the Zeus system received FDA approval the following year [2,3]. Intuitive Surgical Inc. and Computer Motion were both awarded important patents related to their robots, resulting in both going to court to settle an ongoing patent dispute. Before the patent dispute was settled in court, Intuitive Surgical bought out Computer Motion including the disputed patents, thus ending the patent battle. After purchasing Computer Motion, Intuitive Surgical discontinued sale of the Zeus RAS system [3–5].

As a result of Intuitive Surgical's purchase of Computer Motion, many of the patents relating to laparoscopic RAS were owned by Intuitive Surgical up until recently when they began to expire.

Many laparoscopic RAS systems were being developed waiting for the patents to expire, resulting in new robots obtaining FDA approval in the past few years. One of the first to gain FDA approval was the TransEnterix (Morrisville, North Carolina) Senhence RAS system [6]. Senhence is a multi-arm RAS system similar to da Vinci in concept but with some key differences. Unlike the da Vinci RAS system, Senhence uses eye tracking for control of the endoscope, has haptic feedback and individual patient carts each hosting a single robotic arm [7–14]. The CMR Surgical (Cambridge, UK) Versius RAS system is currently awaiting FDA approval, however, has Conformitè Europëenne (CE) approval [6]. Meanwhile some other systems such as the Avatera RAS system by avateramedical GmbH (Jena, Germany) have only obtained CE certification [15]. Meanwhile, other robots have received approval in other countries, such as the REVO-I RAS system by Revo Surgical Solutions (Seoul, Korea) [6]. There are other robots in prototype stages of development undergoing tests in porcine models, cadavers and clinical trials.

This review will look at the RAS systems that are currently commercialized, those that are currently undergoing clinical testing for approval and those in late prototype stages of development. The focus of this review will be robots designed for laparoscopic surgical procedures. This review will take a sub-system approach to comparing and contrasting the RAS systems, beginning with the subsystems of the surgeons console and culminating in the sub-systems of the patient interface.

## 2. Literature Review Methods

Initially, we performed general searches using PubMed Central and Google to collate a list of potential candidate RAS systems for inclusion in the literature review. We then narrowed the inclusion criteria to include only RAS systems that have an endoscope or another internal imaging device for visualization of the surgical environment; have the option to be tele-operated from a remote terminal (i.e., no physical or mechanical connection between the surgeon and the instruments); and the RAS system must be designed for laparoscopic surgery and utilize one or more incision ports through which instruments can maneuver inside the patient's body. Finally, the RAS system must have already been commercialized, or be intended for commercialization in the foreseeable future.

For each included RAS system, a search was performed across eight medically related databases (Cochrane Library, Google Scholar, Ovid, ProQuest Central, PubMED, Science Direct, Scopus and Web of Science) as shown in Table 1. A Boolean search term was used for each RAS system in the format "Robot" AND "Surgery" AND "RAS System Name", for example "Robot" AND "Surgery" AND "da Vinci XI" (Table 1). Where a RAS system has been known by several names, all known names of the system were used in the Boolean search term.

**Table 1.** Literature search results for each robotic-assisted surgery system. Numbers of articles found by each search engine. Results are sorted based on the mean number of publications in each database.

| Manufacturer | Model | Science Direct | Ovid | Web of Science | Scopus | PubMED | Cochrane Library | Google Scholar | ProQuest Central | Mean | Search Term |
|---|---|---|---|---|---|---|---|---|---|---|---|
| avateramedical | Avatera | 4 | 2 | 1 | 1 | 1 | 0 | 22 | 3 | 4 | "Robot" AND "Surgery" AND "Avatera" |
| Intuitive Surgical | da Vinci | 4617 | 2858 | 1899 | 2652 | 1413 | 276 | 26,400 | 2513 | 5329 | "Robot" AND "Surgery" AND "da Vinci" |
| Intuitive Surgical | da Vinci S | 386 | 294 | 120 | 154 | 87 | 22 | 2640 | 302 | 501 | "Robot" AND "Surgery" AND "da Vinci S" |
| Intuitive Surgical | da Vinci SI | 530 | 330 | 90 | 258 | 83 | 40 | 2850 | 358 | 568 | "Robot" AND "Surgery" AND "da Vinci SI" |
| Intuitive Surgical | da Vinci SP | 58 | 32 | 20 | 28 | 13 | 0 | 252 | 14 | 52 | "Robot" AND "Surgery" AND "da Vinci SP" |
| Intuitive Surgical | da Vinci XI | 249 | 136 | 83 | 186 | 78 | 15 | 1530 | 146 | 303 | "Robot" AND "Surgery" AND "da Vinci XI" |
| Medtronic | Hugo | 706 | 106 | 3 | 8 | 32 | 1 | 6320 | 357 | 942 | "Robot" AND "Surgery" AND ("Einstein" OR "Hugo") |
| DLR | MiroSurge | 20 | 8 | 6 | 15 | 1 | 0 | 392 | 34 | 60 | "Robot" AND "Surgery" AND "MiroSurge" |
| Revo Surgical Solutions | Revo | 6 | 11 | 6 | 10 | 6 | 0 | 62 | 9 | 14 | "Robot" AND "Surgery" AND "Revo-I" |
| TransEnterix | Senhance [A] | 51 | 26 | 15 | 28 | 16 | 2 | 335 | 32 | 63 | "Robot" AND "Surgery" AND ("Senhance" OR "ALF-X") |
| Titan Medical | SPORT Surgical System | 19 | 11 | 2 | 4 | 20 | 0 | 130 | 19 | 26 | "Robot" AND "Surgery" AND ("SPORT Surgical System" OR "Single Port Orifice Robotic Technology") |
| ARKANES | SPRINT [C] | 102 | 34 | 5 | 6 | 3 | 0 | 1200 | 45 | 174 | "Robot" AND "Surgery" AND "SPRINT" |
| CMR Surgical [B] | Versius | 8 | 7 | 1 | 7 | 1 | 0 | 70 | 6 | 13 | "Robot" AND "Surgery" AND "Versius" |

(A) Senhance was previously named Telelap ALF-X [8,13]. (B) SPRINT = Single-Port laparoscopy blmaNual robot. (C) CMR Surgical was previously known as Cambridge Medical.

## 3. Literature Review Results

The results of the literature search were collated and presented in Table 1. It was found that the Intuitive Surgical da Vinci RAS systems has had the most number of hits for publications (Table 1). Due to the naming convention of the Intuitive Surgical da Vinci RAS systems, results for newer systems may inflate the number of publications for the first Intuitive Surgical da Vinci RAS system.

The Intuitive Surgical da Vinci RAS system had the most articles written with a mean hit across all databases of 5329 publications. The RAS system with the next highest mean number of publications listed in the databases searched was the Medtronic Hugo RAS system (942 publications).

## 4. The Robotic-Assisted Surgery Systems

The systems selected for this review comprise of RAS systems that have either been commercialized or are intended for commercialization in the near future. The first of these systems is the da Vinci RAS System. The Intuitive Surgical Inc. da Vinci RAS system was the one of the two laparoscopic RAS systems to be introduced in a commercial form to the operating theatre, with the other being the Computer Motion Zeus RAS System.

The da Vinci RAS system set the common format that many RAS systems follow today, comprising of a separate surgeon console and a separate patient cart to house the robotic arms [14,16–19]. The surgeons console on the da Vinci RAS system has the surgeon seated, the surgeon leans into a 3D stereoscopic display to visualize the surgical procedure. The surgeon has two hand controllers and a series of foot pedals through which they can control the robotic arms, instruments and endoscope [14,16–19]. Most recently Intuitive Surgical has introduced the da Vinci SP RAS system which is designed for natural orifice transluminal endoscopic surgery (NOTES) [20].

The Revo Surgical Solutions Revo-I robot and The Avatera RAS system by avateramedical both share a similar configuration to the da Vinci RAS system [15,17,21–26]. Both of these systems have a single patient cart with four arms, a 3D stereoscopic display, hand controls and foot pedals [15,17,21–26]. While the da Vinci RAS system has been in wide commercial use for twenty years, the Revo-I and Avatera RAS systems are only approved and available commercially in limited markets [6,15,26].

The other robots in this review take a different approach to the fore mentioned systems in many aspects. The MiroSurge, Hugo, Senhence, and Versius RAS systems all use flat panel polarized 3D display technology for visualization of the intervention workspace, as opposed to the 3D stereoscopic vision of the previously mentioned systems [6,8,9,11–13,27–44]. Each of these systems also utilize individual patient carts individually housing a single robotic arm [6,8,9,11–13,27–44].

The SPORT Surgical System, SPRINT systems are RAS systems for NOTES, similar to the da Vinci SP system. These robots differ to the da Vinci SP system in that they use a flat panel polarized display. All NOTES systems utilize a single robotic arm that enters the body via a natural orifice. The instruments have additional articulated joints compared to the other RAS systems that enable the instrument the dexterity required of a surgical procedure [33–35,40–42,44,45].

## 5. Imaging and Display Technology

Imaging and display technology are paramount in RAS systems providing the main method of feedback to the surgeon. Before the introduction of tactile and haptic feedback, the imaging and display technology were the sole interface with which a surgeon obtains feedback from the operating environment. Visual feedback ques such as shadows, motion parallax and binocular cues are used to estimate location in 3D space of the end effectors, while tissue deformation is used to estimate gripping and prodding force being applied [46,47].

There are several different approaches taken for display and imaging of the operating workspace for RAS. The imaging technologies consist of two-dimensional (2D) and three-dimensional (3D) endoscopic imaging devices, however now 3D endoscopic devices are exclusively used [13,19,39,48–51]. These can then be coupled to 2D flat panel displays, 3D flat panel displays, 2D stereoscopic displays

and 3D stereoscopic display. Some systems enable the surgeon to choose between 2D and 3D vision [13,19,39,48–51].

3D stereoscopic systems have been in use since the initial commercialization of the Intuitive Systems da Vinci RAS System [16,19,39,48–51]. Three-dimensional stereoscopic systems utilize dual independent displays, one for each eye [16,18,52]. There are several RAS systems that make use of 3D stereoscopic vision for visualizing the operating workspace including all the da Vinci variants, Revo-I and Avatera (Table 2) [15,16,18,19,22,39,48–52]. The screens are placed within close proximity to the eye similar to Virtual Reality headsets. The image is often adjustable by the user either by moving the screens or by adjustment of optical lens to adjust for the individuals' eye spacing [18]. In RAS systems, the 3D stereoscopic vision system is often built into a closed console whereby the surgeon leans into the headset [15,16,18,19,22,39,48–52]. While commercial RAS systems generally utilize a custom 3D stereoscopic vision system, some experimental systems make use of off-the-shelf gaming 3D vision systems such as the Oculus Rift and HTC Vive [19].

The other type of 3D vision often utilized in commercial and experimental RAS systems is 3D polarized flat panel displays. Most systems that do not use 3D stereoscopic vision utilize 3D polarized flat panel vision systems (Table 2), for example Senhance and Versius [6,8,9,11,12,23,26–28,32,34,36,39,47,53–59]. The flat panel displays in these systems are similar to 3D flat panel home televisions, in many cases a commercial variant is actually utilized in these systems. The 3D flat panel displays generally use a HD resolution (1080p) [6,14,16,17,23,28,32,33,43,47,58,60–62]. In order for the surgeon to take advantage of the 3D imaging of the operating workspace, the surgeon must wear a pair of polarized glasses [6,12–14,28–32,34,37–39,43,63]. The flat panel itself displays two images, one for each eye using interlacing [52]. The panel itself is covered in a polarized screen that polarizes the light for each of the horizontal interlaced images at 90 degrees [52]. Each lens of the glasses worn by the surgeon are polarized at 90 degrees to one another such that each eye only sees the image intended for it [52].

There are advantages and disadvantages of each 3D vision system. The 3D stereoscopic system provides a higher fidelity image to the surgeon as each eye has its own screen [52]. This contrasts with polarized 3D displays, whereby the effective horizontal resolution is halved while the vertical resolution is unchanged [52]. The image on 3D stereoscopic systems is also much brighter than the 3D polarized flat screen technology. The polarized lenses in polarized 3D displays eliminate some of the light entering the eye from the screen [26]. The main disadvantage of 3D stereoscopic vision systems is that the surgeons head is buried in the 3D headset isolating the surgeon from the surgical team. Three-dimensional polarized display technology leaves the surgeons head open to the operating theatre minimizing isolation, retaining peripheral vision and allowing for more open communication with the surgical team [26]. One commonality between both display technologies is that they utilize stereoscopic image capture either via a separate computer controlled endoscope or in the case of some single port RAS systems, an integrated stereoscopic camera system [59,64,65].

**Table 2.** Imaging and display technology for each robotic-assisted surgery (RAS) system.

| Robot | 2D | 3D | 3DS | Endoscope Control | Ref. |
|---|---|---|---|---|---|
| Avatera | A | | B | D | [15,26] |
| da Vinci (all versions) | A | | B | D | [16,18,19,39,47–51,66] |
| Hugo | A | C | | N/A | [38] |
| MiroSurge | | C | | N/A | [29,39,63] |
| Revo-I | | | B | N/A | [21,22,24] |
| Senhance | B | C | | ◊ | [8,9,11–13,27,36,37,39,53–57] |
| SPORT Surgical System | | C | | N/A | [33,35,39,42,44] |
| SPRINT | | C | | N/A | [30,31,34,40,41] |
| Versius | | C | | N/A | [6,32,37,39,43] |

2D refers to two-dimensional display. 3D refers to three-dimensional flat panel display.3DS refers to three-dimensional stereoscopic display. N/A refers to information not available at time of publication. A = Secondary 2D flat panel display. B = RAS system contains feature. C = Utilizes 3D polarized glasses. D = Foot switch and hand controls. E = Eye tracking.

## 6. Surgeons Console

As the primary interface between the surgeon and the patient, a well-designed surgeons console is critical for safe surgical procedures using RAS (Figure 1). The console controls must be familiar to the surgeon, easy to operate and have built in safety mechanisms preventing unintentional movement of the end effectors. In addition to a high-quality 3D vision system discussed in the previous section; the console must also be ergonomic such that the surgeon can perform long and complicated surgical procedures with minimal discomfort and fatigue. As with vision systems, there are several different approaches to console design.

### 6.1. Seated or Standing

Traditionally, laparoscopic surgery required the surgeon to stand throughout the surgical procedure at the patient's side, manipulating the laparoscopic instruments while visualizing the internal environment on a monitor. With the introduction of RAS, the surgeon is seated away from the patient to a remote console within the operating theatre, however some new RAS systems have reintroduced the option for the surgeon to stand.

The first major design difference between RAS systems is a sitting or standing posture for the operating surgeon (Table 3, Figure 1). Both sitting and standing offer advantages and disadvantages in the operating theatre. Most RAS systems utilize a seated console for the operating surgeon [8,9,11,14,18,28,33,38,63]. However, Versius allows the surgeon to stand or sit at the robot's console [32,37].

A seated console has many advantages over a standing console. Seated consoles offer less fatigue to the operating surgeon, particularly during long procedures. In the case of Avatera, da Vinci and SPORT, support is offered to the arms by the inclusion of an arm pad, however such support is not offered in Senhence [15,18,44]. Standing consoles allow the surgeon to have a more familiar posture compared to traditional laparoscopic surgery. Additionally, a standing console can feel less isolating to the operating surgeon and the surgical staff, providing a better line of communication. However, as the surgeon must stand while conducting the surgical procedure, the surgeon may experience higher level of fatigue compared to a seated position.

### 6.2. Hand Controllers

As the primary input interface for the surgeon, the hand controllers provide the surgeon with the means to manipulate the position of the end effector in 3D space, in addition to manipulating the end effector itself. The hand controllers must provide maximum dexterity while also being ergonomic so the surgeon can perform long and delicate surgical interventions safely.

A major point of difference between RAS system consoles is the design of the hand controllers (Table 3). When da Vinci was developed, it was designed with controls that attempted to mimic the movement of the end–effectors, rather than emulate the existing laparoscopic instrument controls [11,18]. The fingers are placed in loops of the hand controllers, movement of the thumb and index fingers in a pinching motion control grasping or scissor instruments end effectors [18]. Movement of the hands in three degrees of freedom (DoF) control the rotation of the instruments end effector [18]. This had the advantage of mimicking the motion and grasping of the end–effector, however, it is different from the controls of the traditional laparoscopic instruments. Some studies have shown that the da Vinci hand controllers increased the training time for surgeons used to traditional laparoscopic surgery and may have also increased risk to the patient undergoing an RAS procedure. Other RAS systems (Avatera, MiroSurge, SPORT) use similar hand control interfaces to da Vinci [15,38,42].

Senhence utilizes hand controls that resemble traditional laparoscopic instrumentation controls [8,11,13]. The use of familiar controls was shown to decrease training time for surgeons converting from traditional laparoscopic surgical techniques to RAS. Additionally, it can also help in instances where the surgeon must

transition from using RAS to laparoscopic surgery during a procedure as the controls used are similar. Senhence hand controllers also contain additional buttons, however their utility is unknown [55].

Some newer systems such as the Versius RAS system have controls that are similar to VR gaming controllers [6,12,32,37,39]. The controls feature a hand grip with a looped section in which the index finger is placed for controlling gripping and scissor actions of end effectors [37]. On top of the hand grip is a series of buttons and small joysticks [37]. The joysticks allow the surgeon to adjust the camera position, zoom and rotation [37]. The buttons are used to clutch and declutch the robotic arms and initiate diathermy [37]. The Hugo RAS system uses a hand grip; however, it differs from Versius in that it does not appear to have buttons or joysticks and uses a trigger for grasping/scissoring instead [38].

### 6.3. Haptic Feedback

With traditional laparoscopic surgery, there was a direct physical connection between the surgeon's hand and the end effector allowing the surgeon to 'feel' the end effector and its interaction with the patients tissue. Without this direct physical connection in RAS systems, the surgeon must either rely purely on visual cues or the RAS system needs to provide a method of emulating the physical feedback to the surgeon.

Haptic feedback adds the benefit of force and tactile sensation of the arms and end effectors [67]. Haptic feedback refers many different methods of providing a sensation to the surgeon. Force feedback is a system whereby the force exerted by the end effectors is reflected in a force on the surgeons hands and fingers at the hand manipulators [12,13,26,29,54]. Haptic feedback provides the surgeon with a feel for the force being applied by the instruments to the tissue in the operating environment. It can also provide feeling of the traction and tension the instrument has on the tissue as well as the resistance and slippage of the tissue [8,13,54,68].

Haptic feedback has proven to be challenging to implement. Before feedback can be delivered to the surgeon, the force must be sensed by the RAS system. There are two methods employed for sensing force applied, direct force sensing (DFS) and indirect force sensing (IFS). DFS employs sensors on the instrument tip [69–78]. While this directly measures the forces applied to the tissue, it has the added complexity of requiring the sensors to be small and to be sterilised in systems that have reusable end effectors [6,70,73,74,76–80]. IFS can be achieved by sensors in the robotic arm measuring the force applied by the actuator, or by the computer interoperating visual cues. Since IFS is not integrated into the instrument tip, exposure of the sensors to harsh sterilisation techniques is not of concern [70,76–78,81]. However, as IFS does not directly measure the forces applied by the instrument tip, the haptic feedback delivered to the surgeon can only approximate the actual forces applied [70,76–78,81].

While the da Vinci RAS system does not have haptic feedback, many newer RAS systems have, or are implementing haptic feedback in the form of force feedback (Table 3). With the absence of haptic feedback in the da Vinci RAS system, the surgeon must rely on visual cues to estimate force applied to tissue by the end effectors [54,82,83]. Avatera, MiroSurge, Revo-I, Versius and SPRINT all have haptic feedback, but there is little information available on the implementation for haptic feedback on these RAS systems. The Senhence RAS system includes haptic feedback that provides realistic tactile sensing. Senhence can provide the surgeon with the feeling of force applied by the instruments against tissue [7,8,11–14]. Additionally, the Senhence haptic feedback system can transmit information about the force with which the graspers are grasping tissue and the traction the graspers have on the tissue with 35 grams of sensitivity [7,8,10]. The Senhence system can also amplify the forced sensed by the surgeon, for example during suturing [12,14].

Due to the recency of the introduction of haptic feedback in RAS surgery, most studies into the effectiveness of haptic feedback have been conducted in simulation. However, many studies indicate that haptic feedback to be an advantage in RAS [84–86]. The absence of haptic feedback can result in instances where inappropriate force has been exerted on tissue [79,84]. In addition to reducing harm to patients, haptic feedback may also reduce the learning curve when for adoption of RAS

for surgeons already familiar with laparoscopic surgery [82,84]. Haptics may be more beneficial in learning some tasks such as knot tying, while provide neutral benefit on other tasks such as suturing; when compared to learning without haptic feedback [68,87,88]. However, some other studies suggest the overall learning curve is not affected [9,87].

### 6.4. Tremor Removal

Tremor removal in RAS is where the RAS system removes unwanted natural hand movements transmitted from the surgeon to the instrument. Human hands naturally have a degree of undirected movement, particularly as people age. This movement, if transferred to the instruments during surgery, may pose a risk to the patient. The da Vinci and Senhence RAS systems include tremor removal increasing the precision with which the end effectors can be operated [13,14,16,18]. Other robotic systems may include tremor removal, unfortunately literature on these systems is scarce, hence limited information is available.

### 6.5. Axillary Controls

In addition to the hand controllers RAS systems have some axillary controls for the surgeon to control additional aspects of the RAS system (Table 3). These additional controls which can control things from diathermy to the position and zoom of the endoscope; can take the form of foot pedals, keyboards and touch displays.

In da Vinci, control of the hand manipulators can be switched between endoscope and any of the three instrument arms by the foot pedals [18,19,89]. When the endoscope control foot pedal is depressed, the hand manipulator inputs are diverted to controlling the position of the endoscope. When an instrument arm is not under control of the surgeon, it is locked in position [18,19,89]. The clutch pedal disengages the hand manipulators from all instruments, allowing the surgeon to reposition the hand manipulators [18,19,89]. Tareq, Shahab, Luke and Abhilash [19] suggests that medical errors can be introduced by interruptions to the flow of surgery through the use of foot pedals to switch between instrumentation control and endoscope control. Four other pedals can be configured to activate functions of end-manipulators, such as cauterization [18,19,89]. The Revo-I RAS system has a similar foot pedal control operation to da Vinci [17,24]. Diathermy activation is controlled via the foot pedals on the Hugo RAS system [38]. The Avatera, Hugo and SPORT RAS systems have foot pedals, however while the Avatera RAS system uses foot pedals to control the endoscope, not all the foot pedal functions available on those systems are known [15,38,44].

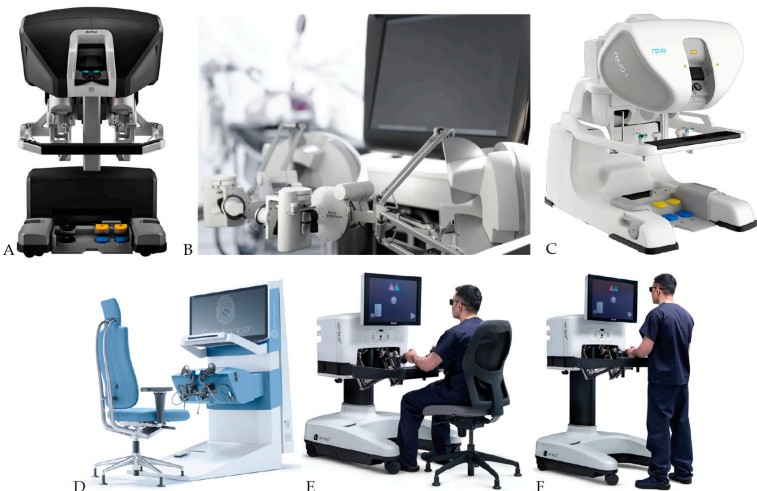

**Figure 1.** Surgeons consoles. (**A**) da Vinci [90], (**B**) MiroSurge [63], (**C**) Revo-I [91] (**D**) Senhence [5], (**E**) Versius seated and (**F**) standing [92].

**Table 3.** Console design and control input devices.

| Robot | Instrument and Arm Control | Instrument Feedback | Tremor Removal | Clutching Arms | Arm Switching | Endoscope Control | Diathermy | Seated or Standing | Reference |
|---|---|---|---|---|---|---|---|---|---|
| Avatera | A | F | N/A | N/A | N/A | H | H | S | [15] |
| da Vinci (all versions) | A | E | P | H | H | H | H | S | [14,16–19] |
| Hugo | B | N/A | N/A | N/A | N/A | N/A | H | S | [38] |
| MiroSurge | A | F | N/A | N/A | N/A | N/A | N/A | S | [28,63,93] |
| Revo-I | A | F | N/A | H | H | H | H | S | [17,22,24] |
| Senhence | C | F | P | N/A | N/A | I | N/A | S | [8,9,11,13,14,26,36,54,56,57,94] |
| SPORT Surgical System | A | N/A | N/A | N/A | NP | N/A | N/A | S | [33,35,42,44] |
| SPRINT | A | N/A | N/A | H | N/A | N/A | N/A | S | [30,31,34,40] |
| Versius | D | F | N/A | G | G | G | G | S/U | [6,12,32,37,39] |

P = Feature is present. NP = Feature is not present. N/A = Information not available at time of writing. A = Manipulator mimics the end effectors via pinching or grasping motion. B = Manipulator is trigger operated. C = Manipulator based on traditional laparoscopic instruments. D = Manipulator resembles a game controller. E = Visual cues are used for instrument feedback. F = Haptic feedback applied to hand controllers. G = Feature provided by axillary controls on hand controller. H = Feature provided by foot pedal. I = Endoscope controlled by eye tracking. S = Seated. U = Seated or standing.

In addition to the foot pedals, other axillary controls are included on many RAS systems; these have many different uses. The da Vinci RAS system includes two additional panels which are located either side of the surgeon allowing adjustment of motion scaling, endoscope calibration as well as system controls such as start, emergency stop and standby [18]. Some later versions of da Vinci include a touch screen display for setting up preferences and operating parameters [58]. The Revo-I RAS system has two emergency stop buttons, one on the right hand side of the surgeons console, and the other on the surgical cart [24]. Senhence include a full size keyboard, however it's utility is unknown [55].

## 7. Patient Interface

The patient interface is the means by which the RAS system interacts with the patient to perform the surgical intervention. This starts with the different approaches to patient carts which host the robotic arms. Some RAS systems have a single patient cart, while some others have an individual cart for each robotic arm. The patient interface then extends to the number of arms available in teach robotic system, the trocar that provides entry into the patient for the robotic arms and the end effectors that interact with the patients' tissue.

### 7.1. Patient Cart

The patient carts in RAS systems hold the robotic arms. Patient carts in RAS systems come in two design styles, individual carts for each instrument arm or a single cart integrating all the instrument arms (Table 4, Figure 2).

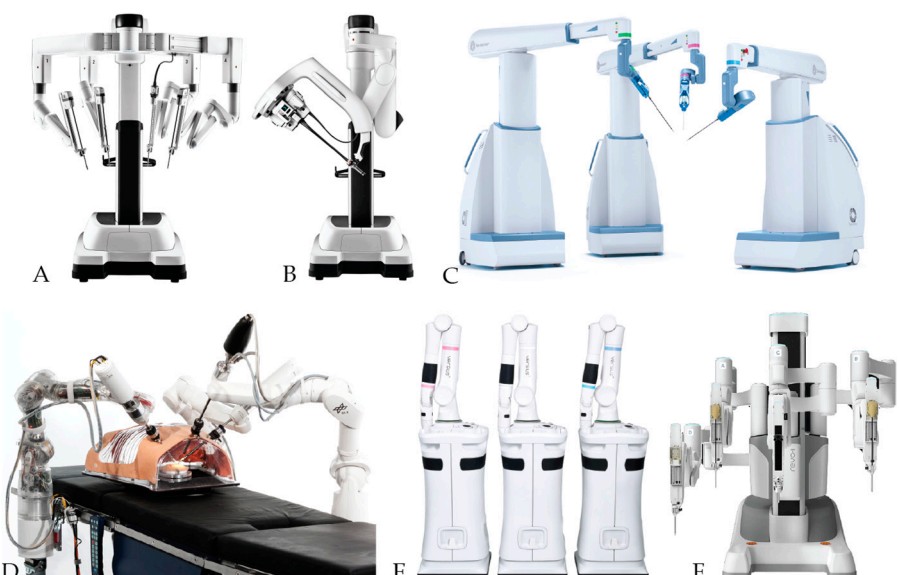

**Figure 2.** Patient carts. (**A**) da Vinci Xi [90], B) da Vinci SP [90], (**C**) Senhence [5], (**D**) MiroSurge [63], (**E**) Versius [92] and (**F**) Revo-I [91].

The Avatera, da Vinci, Revo-I, SPORT Surgical System and SPRINT all use a single integrated cart for all arms [15,18,49]. These carts are mounted on wheels for easy transportation around the operating theatre and between operating theatres [18]. The patient cart is usually placed at the head of the operating table. Most systems with a single integrated cart utilise the central arm for the endoscopic camera, while the other arms hold end effector attachments. The other design for patient carts is to have individual carts for each arm. Hugo, MiroSurge, Senhence and Versius all use individual patient carts for each arm. Like the single integrated cart design, these two are mounted on wheels for easy movability around the operating theatre. However, MiroSurge is not mounted on a wheelable base, but is mounted to the surgical table instead [28,29,63,80].

There are advantages and disadvantages to each approach. With a single integrated patient cart, all the required arms are always located in the one place. The arms have instant registration with each other as they have a common attachment point. With individual patient carts, the carts need to each be registered to the patient, and to each other. Additionally, a single integrated patient cart contains all wiring between the arms within the unit, with connection wires only required to the surgeon's console or central cart. Where individual carts are used, each cart and the surgeons console must all be connected via a central cart which can lead to additional wiring around the operating theatre.

Despite this, individual cart design has some major advantages over a single integrated cart. In single integrated cart solutions, arm clashing can be a problem, as each arm is attached to a common point. This can mean that surgical procedures need to be suspended while arms are re-positioned to avoid an arm clash. In individual cart systems, each individual cart and arm can be positioned such that it provides optimal access to the patient in case of an emergency, while minimising chances of arm clashing [6,13,14,95]. Additionally, some systems that use individual carts such as the Versius and Senhence RAS system, allow the surgical team to use only the number of arms required of the procedure [8,9,11,12,14]. Finally, with individual carts, should an arm fail before or during an intervention, a spare replacement arm can be quickly and easily swapped into place as the arms are interchangeable [8,9]. With integrated single cart systems, the failure of an arm can mean cancellation of a procedure, or the need to revert to manual laparoscopy.

### 7.2. Robot Arms

The robotic arms enable an RAS system to position the end effectors in 3D space within the patient to access the intervention workspace. RAS Systems for laparoscopic surgery traditionally were multiport systems that required a separate robotic arm for each instrument, including the endoscope (Figure 2). However, some newer systems such as the da Vinci SP, SPORT Surgical system and SPRINT are single access port systems [20,40,42,44]. We will discuss differences between multiport and single port systems in more detail in the Access Port and Trocar section below.

Multiport systems were initially commercialized with da Vinci and Zeus in the early 2000's. Both systems were introduced with three robotic arms, two for instruments such as graspers or needle drivers, and one for the endoscope [16,18,49]. However, with newer versions of the da Vinci RAS system, a fourth arm has been introduced [16,18]. Control can be switched between the arms by the surgeon, allowing the surgeon to control all four arms as desired during a surgical intervention [18,19]. Most multiport RAS systems today offer three or four instrument arms and an endoscope arm (Table 4). As discussed previously, some multiport RAS systems such as Senhence are modular and the surgeon can use the number of arms required up to the maximum supported by the system [11,14].

Arms need to be moved into position for insertion of instruments at the beginning and during surgery. The da Vinci RAS system has buttons on each joint that can release the joint allowing the surgical staff to position the arms at the patient. When the button is released, the arm locks back into place [18]. Some other systems such as MiroSurge take a soft robotics approach. In this method, the surgeon or surgical staff can reposition the arms by simply moving them into place. The arms then memorize and maintain the position without the need to use buttons or consoles [29]. Senhence RAS system can detect the trocar and adjust the arm as required for instrument insertion [9].

Single port systems generally have a single arm that docks with the patients access port or trocar [30,31,40,42,44,96]. These systems generally support two instruments plus the endoscope inserted via a guidance tube through the trocar [30,31,40,42,44,96]. In multiport RAS systems there are maximum extents to which robotic arms can move to avoid clashing with other robotic arms. Software used in multiport RAS systems prevents clashing of arms, in some instances can reposition arms to allow full access to the site of intervention, however sometimes physical re-positioning of the arms is sometimes required. Most single port RAS systems avoid this issue by the use of a single robotic arm. The single arm holds all of the instruments required for the intervention which are controlled inside the patient with wires, pulleys and external servo motors [30,31,40,42,44,96].

While Intuitive Surgical has a dedicated single port RAS system in the da Vinci Si, da Vinci Xi can perform single port surgery as well [61,97]. When da Vinci Xi is being used for single port surgery, only two of the three instrument arms are used [61]. The instrument arms cross over in the trocar such that the right instrument arm controls the left end effector, and the left arm the right end effector [61]. The da Vinci controlling computer interprets the hand controls such that the surgeon is unaware of the crossing of the arms [58,61].

Some of the overall movement is provided by the arms, while the rest is provided by the end effectors. Most of the arms in multiport systems provide six to seven degrees of freedom allowing the instruments end effector to move about within the operating workspace. With most single port systems, except for the da Vinci Xi single port solution, the robot arm's DoF is primarily for initial positioning of the insertion tube at the single trocar. Additionally, it provides rotational movement of the entire insertion tube and instruments, or for insertion and retraction to reach deeper into the body.

### 7.3. Trocar

A trocar is the access port or ports inserted into the body of the patient for instruments, endoscopes and gas insufflation during laparoscopic surgery. The port is sealed against the instrument to prevent ingress and egress of fluids, gasses and pathogens during the procedure. The larger the trocar the larger the incision required in the abdomen, which can lead to more prominent scaring. Different sized trocars are used by different multiport robots, with the common sizes of five and eight millimeters for instruments, and up to 12 mm for endoscopes.

In multiport robots, the trocar is used as the fulcrum of the instrument and endoscope [40,66,80]. The instrument pivots around the fulcrum such that movements of the end effector are inverted to the movements of the robotic arm (Figure 3) [7,18,40]. Movements by the external robotic arm are translated through the trocar fulcrum automatically by the RAS system to match the movement of the surgeon [39,98]. Some RAS systems such as Senhence automatically identify the ideal point in the abdominal wall to act as the fulcrum, minimizing movement of the trocar [8,9,13,36]. The MiroSurge has the ability to track the trocar in real time. This enabled MiroSurge to keep the maintain a fixed fulcrum point relative to the trocar even if the trocar is moving. This is critical in procedures such as minimally invasive heart surgery, where the chest wall is constantly moving with respiration. [80].

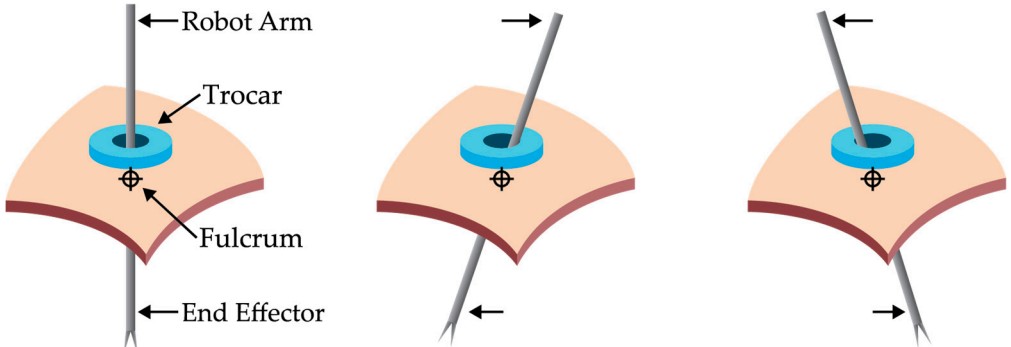

**Figure 3.** Effect of the fulcrum on movement of the end effector by the robot arm. Fulcrum is located within the abdominal wall. When the robot arm moves to the right, the end effector will move left. When the robot arm moves left, the end effector will move right. The motions of the robot arm around the fulcrum are inverted.

In single port RAS, a much larger trocar is required as both the instruments and endoscope must occupy the same port [30,31,40,42,96]. An insertion tube is inserted through the trocar in single port RAS, within which lies internal ports for the instruments and endoscope. The trocars sizes used by the reviewed RAS systems are listed in Table 4. Unlike in multiport robots or traditional laparoscopic surgery, the trocar in single port RAS does not act as a fulcrum for the instruments. The instruments

have joints inside the body that can be manipulated independent of the robotic arm's fulcrum [10,58,61]. The exception is where some multiport systems have been adapted for single port surgery. In these systems the fulcrum is still located in the trocar using a modified instrument [61]. For example, in the case of da Vinci multiarm RAS system single port surgery with the VeSPA instruments (discussed in the next section), the trocar is the point where the robotic arms cross each other as well as the fulcrum for the robotic arm [81].

There are several advantages to single port RAS over multiport. The first point is that it only requires a single incision, therefore the patient suffers less scaring. Additionally, if the natural scar of the navel is used, then scaring can be near invisible [40,96]. Single port RAS has also been shown to leave the patient with less pain compared to multiport approaches [96]. Finally, single port surgery RAS systems pave the way for natural orifice transluminal endoscopic surgery (NOTES). NOTES utilize the natural orifices of the mouth, vagina, urethra or the anus to perform scar-less surgery as the surgical site is accessed via an internal incision where the site of intervention is internal to the abdomen or pelvic cavity [81,99,100].

**Table 4.** Patient Cart and Arm.

| Robot | No. Arms | Instrument Arms [B] | DOF | Trocar | Cart Type | References |
|---|---|---|---|---|---|---|
| Avatera | 4 | 3 | 6 | 5 mm | Single | [15,25,26] |
| da Vinci (except SP) | 4 | 3 | 7 | 8 mm | Single | [16,18,49] |
| da Vinci SP | 1 | 2 | 7 | S 25 mm | Single | [20] |
| Hugo | 4 | 3 | N/A | N/A | Individual | [38] |
| MiroSurge | 3 | 2 | 7 | N/A | Individual [A] | [29,63,82,101] |
| Revo-I | 4 | 3 | 7 | 12 mm | Single | [17,21–24] |
| Senhence | 4 | 3 | 7 | I 5 mm E 10 mm | Individual | [8,11–14,37] |
| SPORT Surgical System | 1 | 2 | N/A | S 25 mm | Single | [42,44,45] |
| SPRINT | 1 | 2 | 6 | S 30 mm | Single | [40,41] |
| Versius | 5 | 4 | 7 | 5 mm | Individual | [6,12,22,37,39,43] |

Single = all arms attached to a single cart; Individual = each arm having its own cart; N/A = information not available at time of writing; DOF = degrees of freedom; A = arms are mounted to the surgical table; B = one arm in each robot is used for the endoscope; E = under trocar column refers to the port size for the endoscope, where as other trocar sizes are for the instruments; S under trocar column refers to single port systems where the instruments and endoscope are inserted through the same trocar.

*7.4. Instruments*

The end effectors are the instruments which are used throughout the operation. These are used to perform incisions, cauterize vascular vessels, suturing, and to manipulate and hold tissue. The end effectors for RAS are similar to manual laparoscopic instruments. Due to being commercially available for twenty years, the da Vinci RAS system has the largest library of end effectors available of all RAS systems (Table 5. Instruments). Not only does the da Vinci RAS system have the largest variety of end effector types, it also has a large variety of each type of end effector. For example, da Vinci has twelve different forceps available for the surgeon to choose from [16,102]. Other RAS systems have a smaller selection of end effectors, and most commonly include types of forceps, graspers, cautery hooks, needle drivers and scissors (Table 2).

Instruments used for cauterizing and coagulation often use diathermy. However, some instruments such as vessel sealers simply use mechanical force to seal a vessel, especially where a vessel needs to be temporarily sealed. Diathermy instruments are either monopolar or bipolar [12,13,15,17,21,23,33,35,39,44,80,102]. Monopolar as the name suggest means that the instrument only contains active electrode of the electrical circuit. A return electrode is attached externally on the patient for the electrical current. The current in monopolar instruments must travel from the instrument, through the body to the return electrode. Bipolar diathermy instruments have both poles within the end effector itself. Bipolar instruments are generally forceps dissectors or graspers whereby the active electrode is on one tip, and return electrodes are on the other. The current only flows between the tips, and not through the patient's body [103,104].

The da Vinci instrument end effectors use their EndoWrist system. The EndoWrist is a driven via cables from servos located in the robotic arm. The proximal end of the instrument contains a series of

reels which connect to the robot arm [18]. These reels are driven by servos located in the robotic arm by meshing drive disks in the instrument with drive discs in the arm. Each reel in the proximal end of the instrument drives cables down the tube, and control one of three DoF in the end effector [18]. The end effector itself has three DoF that reflect the movement of the human wrist. Flexion and extension of the entire end effector; abduction and adduction motion of the tips; and finally open and closing of the end effector tips. The other four out of the seven DoF available in the da Vinci RAS system are provided by the robotic arm, in and out, pitch, yaw and rotation [105].

Information on how instruments function on other multiport RAS systems is sparse, but some information can be garnered from images and video. The Revo-I system surgeon console and patient cart are very similar to da Vinci in both concept and layout. This similarity extends to the instruments which appear to have the same docking mechanism and footprint as the da Vinci RAS system. From images by Abdel Raheem, Troya, Kim, Kim, Won, Joon, Hyun and Rha [21] and in a video by Revo Surgical Solutions [106], it appears that the end effectors function similarly to the da Vinci EndoWrist end effectors. The Verisus RAS system instruments operate similarly to da Vinci, in that they mimic the movement of the wrist (Figure 4).

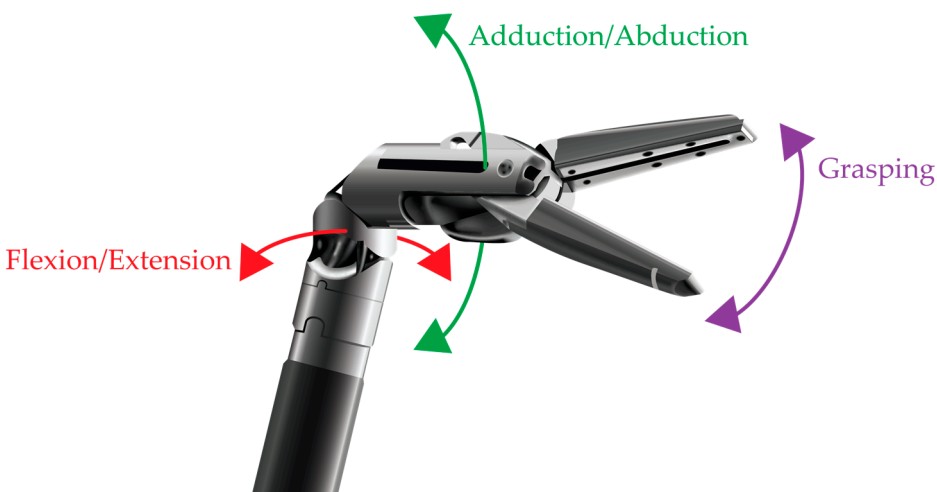

**Figure 4.** The range of motion of the da Vinci EndoWrist end effector. The EndoWrist has motions like the human hand, flexion and extension; adduction and abduction; and grasping.

While the da Vinci EndoWrist instrument has end effectors that have similar function to the human wrist (Figure 4), Senhence instruments are more like traditional laparoscopic instruments [13]. Some Senhence instruments do not have a wrist at the end, but simply have the end effector, for example forceps, at the end of the instrument shaft [107]. While some instruments appear to display some limited flexion and extension movement like da Vinci EndoWrist, but with less range of motion. The instrument seams to rely on shaft rotation and rotation of the tip to achieve a similar range of motion to that of the da Vinci EndoWrist [108,109]. In videos available online by TransEnterix, the end effector can be seen remaining stationary, while the instrument tips open and close cutting and grasping tissue. The robot arms can be seen making required motions to position the end effector at the desired location and orientation as ordered by the surgeon manipulating the robot at the console [107,110].

**Table 5.** Instruments.

| Robot | Cautery Hook | Cautery Spatula | Clip Applier | Dissector | Forceps | Grasper | Needle Drivers | Retractors | Scissors | Sheers | Stapler | Suction/ Irrigator | Vessel Sealer | Reusability | References |
|---|---|---|---|---|---|---|---|---|---|---|---|---|---|---|---|
| Avatera | | | | B | | S | S | | S | | | | | 1 | [15] |
| da Vinci Xi | M | M | S | B | SB | B | S | S | S | SM | S | S | S | ±10 | [16,102] |
| da Vinci SS [1] | M | | S | S | SB | S | S | | S | | | S | | ±10 | [102] |
| da Vinci SP [2] | | | | | S | S | | | S | | | | | N/A | [90] |
| Hugo [3] | | | | | | I | | | | | | | | N/A | [38] |
| MiroSurge | | | | I | I | | S | | I | | | | | N/A | [80,111] |
| Revo-I | M | | I | | I | I | S | | M | | | | | 20 | [17,21,23] |
| Senhence | M | B | | I | | I | | | I | | | | | ∞ | [8,9,11,13,14] |
| SPORT Surgical System | M | | | B | | SB | S | | SM | | | | | 1 | [33,35,42,44] |
| SPRINT [4] | | | | | | I | | | | | | | | N/A | [30,31] |
| Versius | I | I | | | | I | S | | I | | | | | N/A | [12,39] |

1 = SS refers to the da Vinci Xi single site instruments. 2 = Limited information is available on the da Vinci SP instrument suite. 3 = Minimal information available on instruments for the platform at time of writing. S = refers to standard non-electric. B = refers to bipolar diathermy. M = refers to monopolar diathermy. I = Instrument included but further information not available. R = Most instruments can be reused the specified number of times. Some instruments however may have a shorter or longer lifespan. ∞ = There is no hard limit to the number of times an instrument may be reused.

The instruments in single port RAS systems are generally different to those of multiport systems. Rather than utilize individual arms, most RAS systems have a single arm docked at the single trocar. Instruments are inserted through dedicated instrument tubes inside a larger an insertion tube into the abdomen. These instruments do not have external articulation, with all movement occurring inside the abdominal cavity. Most of the instruments use a snake like arm that allows the end effector full range of motion within the operation workspace. The motion can be extended by movement of the single robotic arm. The da Vinci SP system uses different instruments to the other da Vinci systems. The arms for the da Vinci SP feature an elbow joint in the snake like arm in addition to the wrist joint in existing da Vinci EndoWrist instruments. Details on how the elbow joint operations of the da Vinci SP EndoWrist instruments were not available at time of writing. The da Vinci Xi single port instruments are similar to the normal EndoWrist instruments except that the shaft is semi-rigid. This allows the instruments to be inserted through a curved canulae inside the single trocar [61,97].

Another important aspect about RAS instruments is their reusability. The da Vinci RAS system has a strict limit of 10 uses for most instruments (a few instruments have more or less than 10). The instrument itself contains a small printed circuit board that keeps track of the number of times it is used. Once the instrument has reached its end of life, it will no longer function and must be replaced with a fresh instrument [18]. The Revo-I uses a similar system; however, it has a longer working life of 20 reuses [23]. Senhence takes a different approach and does not have a limited lifespan with instruments being replaced as determined by the surgeon [8,9,36]. Reusable instruments must be sterilized between use. Some RAS systems such as Avatera and SPORT Surgical System have disposable instruments and are strictly single use [15,42].

## 8. Future Directions

With competition now in the market for laparoscopic robotic assisted surgery, costs for RAS systems and consumables should start to come down. This in turn should reduce the cost of laparoscopic RAS. Considering the benefits to the patient and reduction in cost, we should see an increase use of RAS for laparoscopic procedures. This will further drive the cost of laparoscopic RAS surgery down as costs of scale reduce outlay for RAS systems, maintenance and consumables.

While da Vinci still has a hold on single port laparoscopic RAS surgery, we can see there are a few competitive systems in late stages of testing. As these systems become available, we should see a similar decrease in cost, and uptake of single port laparoscopic RAS surgery. With the benefits of near scar-less surgery, as costs decrease, single port laparoscopic RAS surgery will likely become the preferred method by surgeons and patients alike. Additionally, hospitals that have purchased the da Vinci Xi system will likely also purchase the single port EndoWrist instruments so that they can perform single-port and multi-port laparoscopic surgery with the one RAS system. With single port laparoscopic RAS systems entering the operating theatre, we will likely see an uptake of NOTES as well for true scar-less surgical procedures. It is likely that as with Intuitive Surgical introduction of a dedicated single port laparoscopic RAS system in the da Vinci SP, they will likely introduce instruments that da Vinci SP can utilize with NOTES procedures to compete with NOTES specialized systems entering the market.

Finally, augmented reality is likely to become a common feature of future RAS systems, and upgrades to existing RAS systems. Augmented reality will enable surgeons to overlay features of the operation workspace on top of live camera feeds from the endoscope [11,112]. This can be used with technologies that can map features such as blood vessels, nerves and even tumours and overlay their location in real time on the surgeon's display [4,113–115]. Additionally, medical imaging previously taken during the diagnosis or planning of an intervention could be overlayed. This will assist surgeons in providing the safest, high quality of care throughout the intervention by helping the surgeon identify the area of interest, while avoiding major blood vessels and nerves that could cause the patient problems post-surgery.

## 9. Conclusions

Robotic-assisted surgery has seen a slow uptake due to cost and the holding of patents by Intuitive Surgical limiting the number of RAS systems in the market. With the expiration of the patents, we are now seeing a rise in the number of new RAS systems available or soon to be available. Several systems have achieved CE certification and are now available in the European Union, while only the TransEnterix Senhence has achieved FDA approval, several others are currently undergoing the process for FDA approval. These new robots will lead to competition and reduce the costs of RAS and will lead to an increase in use. Robotic-assisted surgery will become more common than manual laparoscopic surgery in the near future.

**Author Contributions:** Conceptualization, S.K.L. and G.D.G.; writing—original draft preparation, S.K.L.; writing—review and editing, G.N. and G.D.G.; visualization, S.K.L., supervision, G.D.G. All authors have read and agreed to the published version of the manuscript.

**Funding:** This research was funded by the South Western Institute for Robotics and Automation in Health (SWIRAH).

**Conflicts of Interest:** The authors declare no conflict of interest.

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
