# Peer review of "Laparoscopic Robotic Surgery: Current Perspective and Future Directions"

_robotics, doi:10.3390/robotics9020042_

Round 1

Reviewer 1 Report

The review is well written and gives a good overview over the existing systems. The manuscirpt is well structured with an introduction that leads to the key findings and observations. These are described in an understandable and logic manner.

The article adds to the current knowledge about robot assisted surgery.

Author Response

Thank you for your nice review and comments on our review article

Reviewer 2 Report

The article reviews the state of the art of existing surgical robots designed for laparoscopic surgery, which are commercial or close to the market. The global view is interesting as the study analyses the different robotic systems, their components and functionality, comparing somehow them. The sources of information are well defined so as to justify the systems selected to be described. The compilation and classification of this information is quite useful.

However, the article needs improvements. The article enumerates and compare in a very descriptive way the robots identified and their components. Being so descriptive presents several problems. Reading is boring, and one may lose concepts when going into details of design or operativity without having had a previous global view, mainly for readers that do not know much the field. The way it is presented is more a list of robot architectures, components, features… details, than a clear script following a reasonable thread that starting with more global concepts then evolves towards details. In addition, some concepts are not clear and may induce to misunderstandings.

As an example, the explanation of haptics is confusing. It’s said that Da Vinci has not haptic options, while Avatera, Mirosurge… have. It seems in this aspect they are superior to Da Vinci, but then authors say it’s only used in simulation. Da Vinci has also a simulator. Thus, the previous comparison is not reasonable. There’s no explanation why haptics is not yet solved in current systems (lack of sensors…). It’s justified telling that there is only little information available. This is due to the fact that haptic feedback is not yet solved in teleoperated surgery due to the lack of sensors on the instruments. In simulation is, of course, possible.

Another example of confusing concepts is when dealing with NOTES, line 369: “NOTES….  The surgical site is accessed via an internal incision.”  It’s true that the intervention might need an internal incision, but the NOTES concept implies entering into the body through a natural orifice, as mentioned before. Thus, an internal incision is not necessarily needed.  

Referring to the comparison multiport/single port, the aspect of arms collision should be more justified. The risk is more due to the own robot architecture and the software for arms coordination? Or really due to the fact of being multiport or singleport?. This comparison is also done without considering the kind of interventions. No mention on accessibility or maneuverability with one and the other option.

About paper organization.

After sections 1, 2 and 3 that enumerate existing robots, the sections classification is not very coherent, and the organization is not announced previously. Section 5 is missing, so a section entitled “Imaging and display technology” includes control, pedals, … and so. Then Section 6 “Patient interface” starts with carts, that is, whether the four arms are in the same base structure or if they are independent… and so. The concept of interface here may lead to confusion. To which extend it is referred?

Globally, the paper lacks scientific reasoning, it’s too descriptive and in some points leaves quite open questions to the reader due to lack of justification. The same when dealing with advantages and disadvantages of the different robot options or the described solutions. This also let’s many open questions when data from the literature analyzed is missing.

Some sentences need clarification or justification:

47: “Senhence is a multi-arm RAS 46 system similar to da Vinci in concept but with some key differences.”    Tell which, why…

Many concepts are poorly justified. Let’s mention some:

Typos

17 Patientà patent

 64  maneuverer

99  consider saying displays, visor, viewer …  instead of screen, when the device is not really a screen

150  seated console

154  feels less isolating

156     for the while

174  must transition from

344 to keep the maintain a fixed fulcrum

365  the naval is used

Author Response

Thank you for your feedback, our response is in blue.

The article reviews the state of the art of existing surgical robots designed for laparoscopic surgery, which are commercial or close to the market. The global view is interesting as the study analyses the different robotic systems, their components and functionality, comparing somehow them. The sources of information are well defined so as to justify the systems selected to be described. The compilation and classification of this information is quite useful.

Thank you for your overall feedback on the article.

However, the article needs improvements. The article enumerates and compare in a very descriptive way the robots identified and their components. Being so descriptive presents several problems. Reading is boring, and one may lose concepts when going into details of design or operativity without having had a previous global view, mainly for readers that do not know much the field. The way it is presented is more a list of robot architectures, components, features… details, than a clear script following a reasonable thread that starting with more global concepts then evolves towards details. In addition, some concepts are not clear and may induce to misunderstandings.

Thanks for the comments on the paper. The approach we took was to break down RAS systems into comparable components, rather than into individual RAS systems. We feel that this made the paper easier to follow, and that it was easier to compare and contrast the RAS systems using this method. While we appreciate your comment that one may loose concepts, we feel that this approach does the opposite, since the systems are compared and contrast based on their different components in sections that discuss that particular component, e.g. display technology. This keeps the key concepts about those components together in a single section, rather than the reader having to refer to sections on each individual robot, or refer to a large table.

As an example, the explanation of haptics is confusing. It’s said that Da Vinci has not haptic options, while Avatera, Mirosurge… have. It seems in this aspect they are superior to Da Vinci, but then authors say it’s only used in simulation. Da Vinci has also a simulator. Thus, the previous comparison is not reasonable. There’s no explanation why haptics is not yet solved in current systems (lack of sensors…). It’s justified telling that there is only little information available. This is due to the fact that haptic feedback is not yet solved in teleoperated surgery due to the lack of sensors on the instruments. In simulation is, of course, possible.

Thank you for your feedback on the haptics section. Unfortunately, since the commercialisation of RAS systems incorporating haptics is relatively new, the studies we found were performed in simulation, rather than on actual patients. We would assume that as these new RAS systems that include haptics are more widely adopted, studies will be made into the effectiveness of haptic based RAS systems verses those without haptics such as da Vinci. Because there will be statistics available on safety of the new robots in surgical interventions, which will allow for comparisons between systems with and without haptics. With regards to the lack of haptics in RAS systems until now, one can only speculate as to why Intuitive Surgical has not (and still doesn’t) include haptics in their da Vinci robot.

Another example of confusing concepts is when dealing with NOTES, line 369: “NOTES….  The surgical site is accessed via an internal incision.”  It’s true that the intervention might need an internal incision, but the NOTES concept implies entering into the body through a natural orifice, as mentioned before. Thus, an internal incision is not necessarily needed.

While you are correct in noting that depending on the type of intervention needed, an internal incision may not be required when using NOTES surgery, for example a surgical procedure on the oesophagus or colon. Our paper is targeting laparoscopic robotic surgery, which to our knowledge is generally associated with surgical procedures on the abdomen and pelvic region, and as such generally require an incision through the natural cavity to access the inside of the abdomen/pelvis. We have however extended that sentence to clarify as such.

“NOTES utilises the natural orifices of the mouth, vagina, urethra or the anus to perform scar-less surgery as the surgical site is accessed via an internal incision where the site of intervention is internal to the abdomen or pelvic cavity”

Referring to the comparison multiport/single port, the aspect of arms collision should be more justified. The risk is more due to the own robot architecture and the software for arms coordination? Or really due to the fact of being multiport or singleport?. This comparison is also done without considering the kind of interventions. No mention on accessibility or maneuverability with one and the other option.

Thank you for pointing this out. You are correct in saying that multiport RAS systems avoid clashing through software as well as through placement of the robotic arms. We think the key point is that single port systems negate the need to reposition arms during surgery to access all extents of the intervention location, which may be required with multiport systems.

About paper organization.

After sections 1, 2 and 3 that enumerate existing robots, the sections classification is not very coherent, and the organization is not announced previously. Section 5 is missing, so a section entitled “Imaging and display technology” includes control, pedals, … and so. Then Section 6 “Patient interface” starts with carts, that is, whether the four arms are in the same base structure or if they are independent… and so. The concept of interface here may lead to confusion. To which extend it is referred?

Thanks for your comment, Section 5 is not missing, it is titles 5. Surgeons Console and begins with an introductory paragraph, followed by the subsection titled Seated or Standing. This then continues with the subsections Hand Controllers, Haptic Feedback, Tremor Removal, Foot Pedals and Axillary Controls.

The section titled Patient Interface discusses the parts of the RAS systems that specifically interface with the patient. This section contains subsections titled Patient Cart, Robot Arms, Trocar and Instruments. The definition of interface “a surface regarded as the common boundary of two bodies, spaces, or phases”, we feel sufficiently describes these subsections, as they specifically provide the boundary between the RAS system and the patient, just as the surgeon’s console provides the interface (boundary) between the RAS system and the surgeon. I have however provided an introductory paragraph to the patient interface section to explain what it is and to introduce the sub-sections included.

Globally, the paper lacks scientific reasoning, it’s too descriptive and in some points leaves quite open questions to the reader due to lack of justification. The same when dealing with advantages and disadvantages of the different robot options or the described solutions. This also let’s many open questions when data from the literature analyzed is missing.

Thank you for all of your feedback on the paper. We believe that the paper needs to be descriptive in order to compare and contrast the different approaches to each section of the RAS systems. The paper was intended to present the different robots that have been recently commercialised or intended to be commercialised soon in light of the expiration of many patents held by Intuitive Surgical on the da Vinci RAS system. Therefore we feel that our paper provides a good comparison of the approaches of each robot without being unnecessarily long, providing justification that although RAS surgery has taken time to emerge, it should become more commonplace and affordable in the future.

Reviewer 3 Report

A comprehensive, comparative review of robotic surgery systems, both FDA approved and in the pipeline

Author Response

(The authors gave the same response as above.)

Reviewer 4 Report

  1. This paper was written for comparision of functions and technologies used in different RAS systems in an introduction manner. I think it is more likely be a technique popularization paper. 
  2. If the author add some prographs to introduce related background techniques of each RAS function, the readability will be greatly improved.

Author Response

This paper was written for comparision of functions and technologies used in different RAS systems in an introduction manner. I think it is more likely be a technique popularization paper.

If the author add some prographs to introduce related background techniques of each RAS function, the readability will be greatly improved.

Thank you for your comments. We have updated the paper with paragraphs introducing each function.

Round 2

Reviewer 2 Report

The previous sentences in each section really center the topic and help to smooth the description and main concepts.

Still referring to haptics the lack of force feedback is well known that is due to the lack of effective force sensors to be integrated in the instruments. Miniaturization, need of sterilization, biocompatibility... have difficulted their development. That’s why DaVinci doesn’t provide feedback. Some attempts in extracting force information from visual images have been reported. A reference to that would improve the present version of the article, and justify how the new robots provide this information, or as mentioned, it is only in simulation due to this lack of data.

Author Response

I have added a paragraph based on the feedback from reviewer 2, it has been highlighted by a comment in the uploaded manuscript